# Hypoxia in Lung Cancer Management: A Translational Approach

**DOI:** 10.3390/cancers13143421

**Published:** 2021-07-08

**Authors:** Julien Ancel, Jeanne-Marie Perotin, Maxime Dewolf, Claire Launois, Pauline Mulette, Béatrice Nawrocki-Raby, Véronique Dalstein, Christine Gilles, Gaëtan Deslée, Myriam Polette, Valérian Dormoy

**Affiliations:** 1Inserm UMR-S1250, P3Cell, University of Reims Champagne-Ardenne, SFR CAP-SANTE, 51092 Reims, France; jancel@chu-reims.fr (J.A.); jmperotin-collard@chu-reims.fr (J.-M.P.); pmulette@chu-reims.fr (P.M.); beatrice.raby@univ-reims.fr (B.N.-R.); vdalstein@chu-reims.fr (V.D.); gdeslee@chu-reims.fr (G.D.); myriam.polette@univ-reims.fr (M.P.); 2Department of Respiratory Diseases, Centre Hospitalier Universitaire de Reims, Hôpital Maison Blanche, 51092 Reims, France; mdewolf@chu-reims.fr (M.D.); claunois@chu-reims.fr (C.L.); 3Department of Biopathology, Centre Hospitalier Universitaire de Reims, Hôpital Maison Blanche, 51092 Reims, France; 4Laboratory of Tumor and Development Biology, GIGA-Cancer, University of Liège, 4000 Liège, Belgium; cgilles@uliege.be

**Keywords:** non-small cell lung cancer, hypoxia, HIF, angiogenesis, oxygen sensing, lung cancer management

## Abstract

**Simple Summary:**

Hypoxia is a common feature of lung cancers. Nonetheless, no guidelines have been established to integrate hypoxia-associated biomarkers in patient management. Here, we discuss the current knowledge and provide translational novel considerations regarding its clinical detection and targeting to improve the outcome of patients with non-small-cell lung carcinoma of all stages.

**Abstract:**

Lung cancer represents the first cause of death by cancer worldwide and remains a challenging public health issue. Hypoxia, as a relevant biomarker, has raised high expectations for clinical practice. Here, we review clinical and pathological features related to hypoxic lung tumours. Secondly, we expound on the main current techniques to evaluate hypoxic status in NSCLC focusing on positive emission tomography. We present existing alternative experimental approaches such as the examination of circulating markers and highlight the interest in non-invasive markers. Finally, we evaluate the relevance of investigating hypoxia in lung cancer management as a companion biomarker at various lung cancer stages. Hypoxia could support the identification of patients with higher risks of NSCLC. Moreover, the presence of hypoxia in treated tumours could help clinicians predict a worse prognosis for patients with resected NSCLC and may help identify patients who would benefit potentially from adjuvant therapies. Globally, the large quantity of translational data incites experimental and clinical studies to implement the characterisation of hypoxia in clinical NSCLC management.

## 1. Introduction

Lung cancer represents the leading cause of cancer-related deaths worldwide with 1.761 million deaths in 2018 and an incidence exceeding 2 million (11.6%), largely represented by non-small cell lung cancer (NSCLC) [1]. Lung cancer is diagnosed at a locally advanced or metastatic stage in most cases, leading to poor outcomes and no curative options [2]. In recent decades, many innovative strategies have been designed to improve patient survival rates, namely tyrosine kinase inhibitors (TKIs) of oncogenic alterations or immunotherapies [3]. On the one hand, personalised medicine has emerged from proof of concept to current applications in clinical lung cancer management, but only a limited population harbours molecular targetable alterations and can benefit from this approach [4]. On the other hand, immunotherapies are now largely employed but obtain labile response rates with fewer than 40% of responders among a selected population [5]. Unfortunately, relapse and resistance fatally occur despite specific and adapted strategies. Consequently, global age-standardised 5-year survival remains within the range of 10–20% and a limited increase of up to 5% has been observed [6], arguing the need to further refine and improve clinical lung cancer management.

Hypoxia has been explored specifically in the context of cancer and numerous reports have suggested its potential clinical relevance. The deprivation of optimal oxygen supply at the cellular, tissular, or organ level, is a common feature observed in various physiological and pathological conditions, such as foetal and organ development or interstitial lung disease, chronic obstructive pulmonary disease (COPD), and pulmonary hypertension [7,8,9,10,11,12,13]. Moreover, hypoxia is considered a crucial factor in carcinogenesis [14,15]. Hypoxia appears to be a central key sensor at the forefront of major steps of cancer progression [16] including invasiveness, acquisition of stem cell properties [17], stimulation of angiogenesis and lymphangiogenesis [18], immune escape [19], radiotherapy sensibility, cell survival and resistance to apoptotic signals [20]. Angiogenesis and vasculogenesis are hallmarks of hypoxia-induced modifications [21], but the whole spectrum of molecular and cellular events including oxygen sensing and its signalling pathways are still only partially elucidated [22]. The majority of experimental investigations have been focusing on HIF-1α (hypoxia-inducible factor 1-alpha), the key transcriptional regulator of response to hypoxia [23].

Hypoxia has been identified as a key player in cancer progression and initiation in the context of breast cancer, endocrine tumours, brain tumours, or malignant haematopoiesis processes [24,25,26,27,28]. In this review, we discuss the potential relevance of considering and implementing hypoxia in clinical practice to improve personalised lung cancer management. We focus on NSCLC because of its higher frequency and extensive literature in the field of hypoxia, although some reports also highlight the potential involvement of oxygen starvation in small cell lung cancer [29] and malignant pleural mesothelioma [30].

## 2. Biological Features Associated with Hypoxia in NSCLC

In this first part, we will describe current clinical approaches for evaluating the hypoxic status and its pathological implications in lung cancer specimens. Oxygen electrodes and radionuclide measurements have been used in the past and are still employed despite many drawbacks in their use within clinical settings. Since the initial challenge for the pathologist is the identification of hypoxic tumours in NSCLC, we describe herein the main hypoxia-associated features from sub-cellular to tissue level.

### 2.1. Hypoxia-Inducible Factor Detection in Whole Tumour Tissues

Immunohistochemistry techniques exploring hypoxia usually rely on HIF-1α and pimonidazole staining [31,32]. The expression of HIF-1 inducible markers and the production of nitroimidazoles are triggered by different oxygen concentrations and HIFs can also be induced in physiological hypoxic conditions [33]. Although HIF-1α is most frequently examined to define hypoxic tumours, multi-marker strategies have been implemented previously and based on HIF sub-cellular pathways. In normoxia, the HIF-α subunit is perpetually degraded by the von Hippel-Lindau (VHL) ubiquitination followed by proteasome degradation [34]. Oxygen deprivation inhibits this process and leads to HIF-α stabilisation, an increase of cytosolic concentration, and heterodimerization with β sub-units. Then, the recruitment of co-activators such as CBP (CREB binding protein) or p300 form a transcription complex that bind the promotor region of hypoxia target genes (containing RCGTG sequence especially). Upregulated genes interfere with several biological processes including metabolism (via GLUT-1 (glucose transporter-1) or CA-IX (carbonic anhydrase IX), neo-angiogenesis (via VEGF, vascular endothelial growth factor), epithelial to mesenchymal transition (via vimentin), cancer stem cells (CSC) induction and maintenance (via Sox2), cancer cell migration and invasion (via TGF-α, transforming growth factor-α), and inflammatory mediator production (via IL-6) [35]. HIF-2α and other inducible factors such as Ang-2 (angiopoietin-2), FGF (fibroblast growth factor) or insulin-like growth factors have also been studied and alternative robust hypoxia markers have been evaluated [36,37,38,39,40,41,42,43,44,45]. A meta-analysis, conducted on 17 NSCLC histological studies, extensively described clinical-pathological features associated with hypoxia through HIF-1α and/or VEGF [38]. In this report including 2056 patients, no association was noticed with tumour grade differentiation. Comparing histological sub-types, adenocarcinoma (AC) expressed lower levels of HIF-1α than squamous cell carcinoma (SqCC) (based on tumour positive cells in immunostainings). In a recent study, VEGF-A and Ang-2 were explored in NSCLC [46]. Both AC and SqCC expressed higher levels of VEGF-A and Ang-2 compared to paired-normal tissues. Positive associations between VEGF-A or Ang-2 and tumour size and lymph node metastasis were exclusively observed in AC and not in SqCC. Finally, another meta-analysis based on 30 NSCLC studies reported that HIF-1α expression was higher in the presence of lymph node metastasis (OR = 3.72; *p* < 0.0001), poor degrees of differentiation (OR = 2.12; *p* < 0.0001), or SqCC histology in comparison to AC (OR = 1.28; *p* = 0.03) [47]. In the same meta-analysis, additional expressions of hypoxia-related markers such as VEGF or CA-IX were assessed, with similar results to HIF-1α. Hypoxic features were also associated with higher tumoural microvessel density [48,49]. Although additional data are needed, several reports focused on HIF-2α expression in NSCLC with similar associations to HIF-1α [50]. Hypoxia could thus support carcinogenesis and tumour progression in the two histological sub-types involving distinct molecular pathways [51,52,53], ultimately leading to differential hypoxic-adaptation between AC and SqCC. Such carcinogenesis mechanisms could be supported by tumour-initiating cells.

### 2.2. Tumour-Initiating Cells and Hypoxic Conditions

#### 2.2.1. Cancer Stem Cells Are Influenced by Hypoxia

Evidence is growing that some tumour cells harbour and retain a tumour-initiating cell phenotype involved in tumourigenesis or therapeutic resistance [38,54,55]. Interestingly, such CSC were preferentially observed in hypoxic areas, named niches, and both HIF-1α and HIF-2α were associated with induction and/or maintenance of CSCs across many cancer types including lung cancer [56]. CSC emerged as a promising model for cancer kinetic analysis from tumour initiation to invasion and metastasis [57]. Hypoxic-condition cultures of human primary lung cancer cells resulted in an enrichment of CSC harbouring aggressive and invasiveness features [58], suggesting selective pressure induced by hypoxia. Furthermore, hypoxia demonstrated its capacity to induce CSC properties, potentially supported by CXCR4 (C-X-C motif chemokine receptor 4) activation [59]. Invasion and metastatic properties can also be promoted by hypoxia through CSC stimulation [60]. Additionally, hypoxia-conditioned CSCs were associated with lower responses to therapies and poorer prognosis [61].

#### 2.2.2. Epithelial to Mesenchymal Polarization by Hypoxia

Well in line with the observations that hypoxia enhances metastatic properties in various cancers and with important overlaps between EMT (epithelial–mesenchymal transition) and CSC features, a close relationship between EMT and hypoxic condition was also established, including mutual cell signalling pathways [62] or epigenetic regulations [63]. In the context of lung cancer, mesenchymal markers such as vimentin, and EMT transcription factors such as TWIST1 (Twist family BHLH transcription factor 1) or Snail were found overexpressed in hypoxic lung specimens while epithelial markers such as E-cadherin were repressed compared to non-hypoxic samples [64,65,66,67,68,69]. Such close relationships between EMT and hypoxia could thus be involved in aggressive phenotypes and resistance sustained by mesenchymal polarization [70,71,72]. Considering more frequent hypoxic features in mesenchymal tumours and higher aggressiveness and metastatic properties allocated to non-epithelial specimens, hypoxia appears as an important mechanism supporting therapy resistance such as chemoresistance in hypoxic tumours. Interestingly, hypoxia-induced features such as EMT appear reversible, offering opportunities for drug development targeting hypoxic related-mechanisms [60,73,74].

### 2.3. Tumour Microenvironment Features in Hypoxic Condition

Along with the establishment of histological sub-types, additional phenotypical and immuno-histological observations are essential to refine the clinical strategy. The hypoxic condition impacts the entire tumour microenvironment (TME), contributing to the complex interplays established between tumour cells and neighbouring populations [75].

#### 2.3.1. Hypoxia-Driven Stroma Modifications

Hypoxia impacts cancer-associated fibroblasts (CAF), immune cells, endothelial cells [76,77], or mesenchymal stem cells (MSCs) [78]. MSCs preconditioned via oxygen deprivation stimulated EMT which resulted in both enhanced properties of aggressiveness for tumour cells and M2 polarization of tumour-associated macrophages [79]. Exosomes, corresponding to 30–150 nm extracellular vesicles from the endosomal compartment, appeared to be largely involved in the molecular mechanisms of this intercellular dialogue between tumour cells and their distant microenvironment, as they could carry various sub-cellular information such as proteins, nucleic acids, or metabolites [78,80,81,82,83]. Hypoxia-activated host cells or hypoxia-driven stroma modifications were shown to sustain tumour cell aggressiveness and CSC/EMT features [84]. Moreover, hypoxia-induced tumour microenvironment morphological and biological modifications are associated with drug resistance and could also represent potential and promising aspects to target [85,86,87,88].

#### 2.3.2. Inflammation Landscape in the Hypoxic Context

Hypoxia has proven to be a crucial regulator of inflammation and of many immune cell functions that are key targets of therapeutic strategies aiming at either restoring and/or activating anti-tumour immunity. Important crosstalk and bidirectional induction between HIFs and NF-κB (nuclear factor kappa-light-chain-enhancer of activated B cells), a potent inducer of inflammation, were thus reported in various cancers including NSCLC [89].

Immune cells of the tumour microenvironment also play a crucial role in cancer development and immune response through tumour-associated macrophages (TAMs) or tumour-infiltrating lymphocytes (TILs) [90,91]. TAMs, representing most of the leukocyte population in the TME of solid malignancies, particularly mediate pro and/or anti-neoplastic effects dependent on macrophage polarization [92]. More particularly, they enhance tumour hypoxia and aerobic glycolysis in NSCLC both in pre-clinical and clinical contexts [93]. TAM depletion by clodronate was shown to reduce tumour hypoxia and increase intra-tumoural T-cell infiltrations, restoring efficacy to immune checkpoint inhibitor (ICI) [94]. The interactions between TAMs and hypoxia seem particularly promising and original therapeutic approaches are in development [95]. Moreover, hypoxia strongly impacts TILs in lung cancer [96]. A prognostic determinant 8-gene signature, based on TILs and cell oxygen sensing in 2712 tumours (including 515 NSCLC), was identified [97]. These reports are consistent with other markers and cohorts that establish positive correlations between hypoxic markers and TILs profusion and quiescence [98,99,100,101], highlighting a strong contribution of hypoxia in tumoural and peri-tumoural immune features.

#### 2.3.3. Immune Checkpoint Disruption by Hypoxia

Hypoxia was also shown to induce PD-L1 expression to orchestrate cancer immune escape [102,103,104]. Immunotherapies and ICI such as anti-cytotoxic T-lymphocyte antigen-4 (CTLA-4) or anti-programmed death-ligand 1/programmed death-1 (PD-L1/PD-1) provided a significant improvement in NSCLC management [105]. Anti PD-L1 interact preferentially with PD-L1 at the surface of immune cells to restore the anti-tumoural immune response [106]. A significant correlation between PD-L1 and HIF-1α, VEGF-1 and CA-IX restricted to AC was reported in a cohort of 295 NSCLC [104].

### 2.4. Molecular Signature of Hypoxic Tumours

Additional and complemental consequences driven by hypoxia can be observed in NSCLC tumours, including metabolic reprogramming, especially with aerobic glycolysis upregulation, exosome biogenesis and secretion, neo-angiogenesis, lymphangiogenesis, or reactive oxygen species (ROS) production [84,85,86]. The molecular modifications serve as readouts of the cellular hypoxia-induced alterations.

#### 2.4.1. Metabolic Consequences of Hypoxia

Aerobic glycolysis has been described as an adaptive mechanism of tumour growth in hypoxic conditions [107]. This persistent metabolism to lactate from glucose acidifies the peritumoural microenvironment and induces subpopulation cell resistance with enhanced division, and growth capacity. Interestingly, lung cancer cells and their microenvironment were investigated at the level of the cellular metabolisms such as lactate or pyruvate dehydrogenases (LDH and PDK, respectively), glucose transporters (GLUT), and HIF-1α/2α. In this comparative metabolic profile, cancer cells exhibited anaerobic features with higher GLUT capacities or lactate extrusion while TAMs presented a complementary profile with aerobic implications and lactate oxidation [108]. This report described a close interplay between the tumour and its peritumoural environment, especially regarding aerobic glycolysis, and highlighted additional molecular targets to evaluate hypoxia-induced cellular alterations.

#### 2.4.2. Hypoxia Widely Impairs Cancer Gene Expression

High-throughput multi-omic approaches established the genetic prints of tumour hypoxia, introducing both complexities and clinical relevance to consider hypoxia in cancer. An impaired non-supervised pan-cancer core-metabolic gene signature was strongly associated with hypoxia in most common types of cancer [109]. Many pan-cancer (including NSCLC) hypoxia-associated gene expression signatures have been published and associated with a poor outcome [110,111,112,113,114].

#### 2.4.3. Hypoxia Supports Molecular Alterations

A finely-tuned understanding of cancer biomolecular alterations led to major improvements for patient management as illustrated by the targeting of EGFR (epidermal growth factor receptor) alterations or ALK (anaplastic lymphoma kinase)/ROS1 (ROS proto-oncogene 1) rearrangements in NSCLC [115]. Hypoxia might be involved in the incidence of these molecular alterations: it was associated with high genomic instability and somatic genomic changes in a cancer report of 19 various tumour types. Hypoxic tumours exhibited characteristic driver-mutation signatures [116] and higher tumour mutational load with higher frequencies of driver mutations in PTEN (phosphatase and TENsin homolog), MYC (MYC proto-oncogene, BHLH transcription factor) and TP53 (tumour protein p53) [117]. Despite a few controversies, molecular polymorphisms have been extensively explored and HIF [118,119] or VEGF [120,121] polymorphisms have been associated with poor prognosis.

#### 2.4.4. EGFR and ALK Genes Are More Frequently Altered in Hypoxic NSCLC Tumours

EGFR is an established activator of the HIF pathway, contributing to tumour progression. Data are also consistent with antiangiogenic effects in EGFR-TKIs in addition to proapoptotic activity. Hypoxia is proposed as a mechanism of resistance to anti-EGFR therapies that could be reversed [122,123,124,125,126]. For example, Yuan et al. explored associations between angiogenesis-inducible factor expressions and the presence of EGFR and KRAS (Kirsten ras oncogene homolog) mutations to specify the relationship between oncogenic alterations and hypoxia [127]. EGFR mutations concerning 21 or 20 exons (*p* = 0.002) were associated with significantly high levels of VEGF-A. In contrast, tumours harbouring an exon 19 mutated EGFR exhibited lower levels of VEGFR1 (vascular endothelial growth factor receptor 1). KRAS mutations were also positively associated with VEGF/VEGFRs expression. Molecular cooperation between HIF-1α and c-Jun (c-Jun proto-oncogene) was described in NSCLC tumours harbouring an activating mutation of EGFR, resulting in both primary and acquired resistance to EGFR-TKI [128]. In another model, NSCLC cell lines with EGFR alterations (including T790M mutation, sensitizing to Osimertinib) were exposed to Osimertinib with a chronic and moderate hypoxic atmosphere. Interestingly, hypoxic NSCLC cell lines developed a resistance associated with a mesenchymal polarization and supported by FGFR1 (fibroblast growth factor receptor 1), largely resulting from mitogen-activated protein kinases (MAPK) pathway activation [129].

To summarise, hypoxia, and its cellular and molecular alterations, orchestrates hallmarks of lung cancer progression and metastasis such as CSC induction and maintenance, EMT, TME remodelling, immune escape, or oncogenic driver acquisition. This suggests a potentially promising clinical relevance for hypoxia characterisation in NSCLC. Figure 1 summarises cyto-histological features observed in hypoxic NSCLC tumours during lung carcinogenesis. There are currently two main limitations to integrating hypoxia status in real-time lung cancer management. Firstly, hypoxia can induce heterogeneous effects [130] after acute or chronic exposure. Cyclic levels of oxygen contribute to the hypoxia-induced cellular and molecular alterations [131]. Second, hypoxic markers rely on pathological examination and are limited due to tumour heterogeneity [132,133,134] and re-biopsy feasibility [135].

## 3. Available Tools to Detect Hypoxia in Clinical Practice

Innovative approaches are needed to tackle current clinical challenges in lung cancer management and to overcome current drawbacks. Clinicians appeal for easy-to-use robust markers to overcome tumour heterogeneity, stage cancer disease, detect metastasis, and predict patient outcomes during and after treatment. Thus, we review here alternative non-invasive methods evaluating hypoxia in NSCLC, and we highlight their potential relevance to help clinicians identify hypoxic tumours. These complementary approaches are depicted in Figure 2. Table 1 focuses on hypoxia-related markers with potential clinical relevance, considering their interest and respective limitations at meeting clinicians’ expectations.

### 3.1. Hypoxic Characterisation by Imaging Techniques

Various non-invasive methods are available for clinicians to assess hypoxia in solid cancer cases. Lung cancer management currently involves contrast-enhanced computed tomography (CT) and positron emission tomography (PET) and could provide insight into hypoxic features. Other options include ultrasound imagery and magnetic resonance imaging which have been employed in clinical studies and might be relevant to assess hypoxia, especially in head and neck carcinoma [149,150,151]. However, they are not currently employed in lung cancer and hardly transposable for thoracic malignity.

#### 3.1.1. Using Radiomics on Computed Tomography Images to Identify Hypoxic Tumours

CT is usually the first modality of imaging implemented in lung cancer management to establish a preliminary endo-thoracic staging. The emergence of radiomics in oncology unveiled the potential to analyse and quantify sub-visual features of CT images such as texture, wavelet, structure, and intensity to correlate with pathological characteristics [152]. Radiomics consists of the automatically high throughput extraction of a large amount of quantitative data from CT acquisitions and establishing models which predict bio-pathological features through a non-invasive approach [153]. A study by Ganeshan et al. sought to assess hypoxia by radiomics in 14 patients who underwent lung cancer surgery after intravenous injection of pimonidazole, a pathological hypoxia-related marker [154]. Pathological findings and image acquisition were then matched and compared. Two radiomic parameters (standard deviation of all pixel values and mean value of positive pixels) demonstrated the value of texture parameters. They correlated with hypoxic- and angiogenesis-related features observed by pathologists (i.e., angiogenesis explored by CD34 expression or pimonidazole staining). Other CT parameters based on enhancement after contrast injection, such as blood volume and flow-extraction products, also correlated with hypoxic histological findings in surgically resected NSCLC cohorts [139,155]. Additionally, radiomics and radiogenomics may help classify tumours based on their hypoxia status: radiomic features analysed with artificial intelligence were correlated with tumour hypoxia and further distinguished hypoxic area inside tumour heterogeneity [156]. CT texture analysis is confronted with non-standardised methods of acquisition, and further research is needed to establish guidelines [157] despite interesting results reported in NSCLC management.

#### 3.1.2. Positive Emission Tomography Radiotracers Identify Hypoxia

Image acquisition by PET (based on cell metabolism activity) might be a promising non-invasive modality to better characterise hypoxia in NSCLC. Many PET tracers have been designed to identify the differential between hypoxic regions within tumours. They are mainly engineered based on a 2-nitroimidazole structure including [^18^F]-HX4 (3-[^18^F]fluoro-2-(4-((2-nitro-1Himidazol-1-yl)methyl)-1H-1,2,3,-triazol-1-yl)-propan-1-ol) or [^18^F]-FAZA (1-(5-[^18^F]Fluoro-5-deoxy-α-D-arabinofuranosyl)-2-nitroimidazole) [158]. The specific features for each radiotracer are not discussed here. Among them, [^18^F]-MISO (^18^F-fluoromisonidazole) has been identified as a robust marker of hypoxia [159,160] with good reproducibility [161] and validity [162,163,164], highlighting [^18^F]-MISO as a potential hypoxia-relevant biomarker, especially in radiation management [165]. [^18^F]-HX4 PET imaging also seems promising to document hypoxia in NSCLC [166,167,168]. [^18^F]-HX4 and [^18^F]-MISO successfully identified hypoxic areas within patient tumours four hours after intravenous injection. Interestingly, [^18^F]-HX4 may emerge as the best option since it presents a higher contrast than [^18^F]-MISO [169,170]. Additional radiotracers have been tested in preclinical and clinical models of NSCLC [166,167,168,171] but will not be discussed here.

#### 3.1.3. Conventional [^18^F]-FDG PET to Explore Hypoxia

Specific hypoxia radiotracers are expensive and currently not available in clinical routine. So far, PET imaging employs tracers composed of [^18^F]fluorodeoxyglucose (FDG), allowing clinicians to stage tumoural disease and identify potential lymph nodes and visceral metastasis [172]. Hypoxic examination by [^18^F]-FDG PET has also been explored in NSCLC. A robust positive correlation between standardised uptake values (SUV) derived from [^18^F]-FDG and HIF-1α expression based on three studies including 288 patients was addressed in a meta-analysis (pooled correlation of 0.42; *p* = 0.02). Additionally, five reports based on 310 patients identified a positive correlation between SUV and microvessel density [140]. [^18^F]-FDG uptake and hypoxia association (r = 0.54; *p* < 0.0001) were also concordant with other reports exploring different approaches. A genetic signature associated with SUV intensity in NSCLC and largely consistent with hypoxia sensing was described and validated [173]. Since CD68 TAMs in the tumoural microenvironment are influenced by hypoxia, [^18^F]-FDG PET has demonstrated its capacity to predict hypoxia in vivo with a positive correlation between SUV and CD68 positive staining, as reported in 98 matched NSCLC specimens [94].

#### 3.1.4. Current Limitations for Hypoxia Characterisation by PET

Whether [^18^F]-FDG and [^18^F]-MISO can be useful, to characterise hypoxia requires additional investigations to assess their complementarity and limits. The efficiency of PET radiotracers to identify hypoxia could be evaluated on the tumour-muscle ratio in the same post-injection time lapse in the same population. However, comparisons are difficult in the absence of standardised methods. Other parameters such as tumour-to-blood ratios might be promising in hypoxic region determination [168]. Different kinetics and uptake profiles were obtained in a comparison of 34 non-resectable NSCLC hypermetabolic and hypoxic volumes, investigated by [^18^F]-FDG and [^18^F]-MISO [174]. To date, [^18^F]-FDG PET based on SUV is used in daily clinical practice to determine malignancy probability before histologic confirmation or to perform a whole-body staging. As previously exposed, standard SUV acquisition by [^18^F]-FDG PET alone appears insufficient to characterise hypoxic tumour regions with mismatch between metabolic and hypoxic regions. Other hypoxic-related radiotracers are thus needed and their addition to [^18^F]-FDG tracer could be interesting. However, novel hypoxic-tracers’ validity, reproducibility, specificity, and standardisation represent the main current limitations for hypoxic investigation by PET.

In terms of accuracy, PET radiotracers are promising as non-invasive procedures. The [^18^F]-HX4 tracer seems the most promising in the hypoxia field but the signal-to-noise ratio requires improvement [175]. Larger cohorts are needed to establish clinical relevance, by confronting various PET-acquired parameters with pathologically-matched hypoxia features and association with prognostics. All in all, hypoxia imaging by PET acquisition offers many advantages beyond its non-invasive interest. Complementary radiotracers can be simultaneously used to explore patient tumour features [176]. Their associations could overcome current limits to their daily use in clinical management. Combining such non-invasive approaches could improve the clinical relevance of PET/CT imaging as illustrated by the growing interest for radiomic analyses of PET in NSCLC [177] or multimodal imaging to concomitantly assess perfusion, metabolism and hypoxia [178]. Although PET and CT are appealing procedures with a strong potential to evaluate hypoxia in patient tumours, their implementation in a reasonable real-time monitoring procedure is subjected to the availability of equipment, costs, and radiation exposition. [^18^F]-HX4 has appeared as the most promising tracer but clinical validation of alternative candidates is required to establish experimental guidelines (i.e., perfusion, acquisition parameters, etc.). Multimodal characterisation broadly based on blood-derived cancer products could overcome PET-associated drawbacks.

### 3.2. Circulating Markers to Help Clinicians Classify Hypoxic Tumours

Blood markers appear as a promising approach to meet clinical expectations. They avoid iterative radiation exposition, and technical bias induced by various methods of acquisition and analysis. Blood analysis is standardised, presents very low inter-operator biases, and reports circulating parameters that adapt therapeutics in a real-time manner. Several biomarkers that are assessable in blood could be of interest in the context of hypoxia: namely soluble protein, and derived tumour products such as tumour circulating DNA, circulating tumour cells, or circulating exosomes.

#### 3.2.1. Soluble Blood Proteins Are Promising for the Identification of Hypoxic Tumours

So far, hypoxia investigations have focused mainly on serum blood biomarkers assessment. In a lung cancer cohort enrolling 263 non-metastatic and non-operable NSCLC treated by radiotherapy alone or with chemotherapy, high OPN and CA-IX serum concentrations (two blood markers related to the hypoxia pathway) were identified as negatively impacting overall survival (OS) [141]. The prognostic impact of various serum hypoxia-related proteins, such as OPN, CA-IX, and VEGF, was evaluated in a pilot study of 55 non-metastatic NSCLC undergoing radical radiotherapy [138]. They all correlated with clinical parameters associated with OS and a combined consideration of expression for those three markers was independently associated with poor outcome. Validation was then obtained in larger NSCLC cohorts, especially focused on OPN plasma levels [142,143,179]. HIF-1α serum levels in 80 patients harbouring NSCLC and undergoing chemoradiotherapy in a curative intent were assessed [137]: soluble HIF-1α levels were decreased during and after treatment compared to the initial point, suggesting serum HIF-1α concentrations as a potential marker to monitor tumour response during chemoradiation. Other soluble hypoxia-related biomarkers were assessed in patient serum with NSCLC including VEGF and basic fibroblast growth factor (bFGF): similar results were reported with poorer survival rates among patients with high serum concentrations of circulating VEGF and bFGF [147]. Despite interesting results that predict poor clinical outcomes for numerous hypoxia-related soluble markers, and their potential for the monitoring and management of lung cancer patients, a validation in larger cohorts is needed to prospectively confirm their clinical relevance and provide guidelines on blood measurements. Available reports were restricted to NSCLC treated by radiation, and other treatment modalities require exploration.

#### 3.2.2. Tumour Circulating DNA/RNA Provide Hypoxia-Related Signatures

Tumour-released circulating DNA detection was recently implemented in the clinical practice of NSCLC, especially in the field of oncogenic drivers [180]. The interrelation between circulating tumour products and hypoxia condition was investigated. Hypoxia was associated with an increase of both circulating tumour DNA and tumour growth in a model of mice grafted with lung cancer cell lines and exposed to intermittent hypoxia [181,182]. Circulating miRNA levels, a sub-group of small noncoding RNAs mainly repressing post-transcriptional gene expression, were also investigated in the context of hypoxia. Depending on the tumoural context, including hypoxia [183], they exerted both pro and/or anti oncogenic properties. Circulating miRNA (miR)-21, miR-128, miR-155, and miR-181a were quantified in 128 NSCLC cases undergoing first-line platinum-based chemotherapy treatment [148]. Higher expressions of miRNA levels were associated with worse survival. Integrated pathway analysis highlighted significant hypoxia-signalling involvement in miRNA targets. miRNAs present a great potential both as biomarkers and as a therapeutic strategy. However, their molecular and cellular mechanisms are not completely elucidated, and their assessment requires standardisation.

#### 3.2.3. Circulating Tumour Cells Associated with Hypoxic Tumours

Among other tumour-circulating products, CTCs are reported in solid cancers but are still not currently used in clinical management [184]. In vitro evidence suggesting an interplay between hypoxia and CTC is growing. For instance, tumour-harbouring mesenchymal traits were characterised by EMT induction and mesenchymal polarisation in hypoxic conditions concomitant with an increased CTC count [36]. Additionally, hypoxia-inducing cancer stem cells and tumour growth might be associated with the emergence of CTCs [84]. In clinical conditions, CTCs were used to discriminate hypoxic status in metastatic breast cancers based on their HIF-1α and VEGF expressions with potential relevance [185]. Although CTCs were already used in NSCLC to assess features such as PD-L1 expression and/or EMT [186,187], no reports on hypoxia characterisation are currently available in lung cancer.

### 3.3. Emerging Approaches in Hypoxic-Tumour Identification

Circulating tumour DNA methylation might represent a promising and innovative approach to identifying markers able to deal with different challenges in the cancer field such as prediction of cancer risk [188]. Oxygen deprivation broadly impacts histone methylation in various types of cancer, including NSCLC [189,190,191] and cancer stem cells [59]. These epigenetic DNA modifications are clearly described in hypoxic conditions [192,193] and additional molecular targets are explored such as collagen 1A1 (COL1A1), RNA-binding and pre-mRNA processing factor 40 homolog A (PRPF40A) or uncoupling protein 2 (UCP2) [194,195]. The methylation status of over 1800 genes was altered in hypoxic conditions in cases of lung adenocarcinoma. Pathway enrichment analysis identified a large involvement in metabolic signalling, angiogenesis and cancer progression with a special interest for FAM20C (family with sequence similarity 20, member C), MYLIP (myosin regulatory light chain interacting protein), and COL7A1 (collagen type VII alpha 1 chain) predicting worsening outcomes for patients [196]. A close relationship between hypoxia (based on HIF-1α expression) and EGLN2 (Egl nine homolog 2) methylation was observed in lung adenocarcinoma and SqCC (*n* = 1230). EGLN2 methylation was associated with hypoxic features while the association of HIF-1α and EGLN2 enabled the identification of patients with poorer OS [197]. Other targets linked to hypoxia such as Cdkn2a (cyclin-dependent kinase inhibitor 2A) or Vdac1 (voltage-dependent anion-selective channel 1) [198,199] were identified in an innovative approach aiming to explore novel biomarkers for lung cancer in exhaled breath [200,201].

Innovative imaging procedures are also in development. The partial elucidation of the spatio-temporal signature of oxygen deprivation introduced an essential component of hypoxia. Experimental investigations are required to understand the impact of cyclic hypoxia in cancer genesis and progression [131]. The distribution and severity of hypoxia evaluated by the quantitative geographic monitoring of oxygen concentrations could bring additional relevant information to break current clinical limitations and modify therapy outcomes [202]. Other approaches such as metabolomics or proteomic analysis could represent interesting techniques [111,203,204,205,206]. Such emerging technologies offer wide perspectives, but they require validation regarding their contribution to hypoxia.

## 4. Prognostic Implications of Hypoxia in Lung Cancer

NSCLC presents specific challenges depending on its stage of development. Locally advanced NSCLC, mostly represented by stages III of the IASLC classification, can still be cured but requires robust predictors of response to radiotherapy and chemoradiotherapy. Locally advanced stages share a mutual challenge, based on markers that can predict recurrence risk and outcome after curative treatment. Unfortunately, metastatic conditions remain common, and treatments sequentially based on chemotherapies, immunotherapies, and targeted therapies would also benefit from additional biomarkers capable of predicting tumour response.

### 4.1. Early and Locally Advanced Stages

#### 4.1.1. Hypoxia Is Associated with Higher Tumour Stages

Accurate staging is a cornerstone in lung cancer management. Visceral metastasis detection leads to no-curative intent while lymph node metastases impose a more aggressive strategy combining (neo-)adjuvant therapies. Current NSCLC staging largely relies on imaging and mainly on PET/CT acquisition, which remains a transversal acquisition. Hypoxia could complement clinical staging and bring additional informative data on patient staging.

HIF-1α potential as a marker to identify patients with lymph node metastasis was investigated. Concordant with the association between HIF-1α expression and lymph node metastasis, high mRNA HIF-1α expression was also associated with tumour size and pathological stage (*p* < 0.05) [207]. HIF-1-induced products such as VEGF-A were also positively associated with larger tumour size, presence of lymph node metastasis, and more advanced pathological stages [48,49,208]. Additionally, other VEGFs proteins related to lymphangiogenesis such as VEGF-C were also positively associated with lymph node metastasis [209,210,211,212].

#### 4.1.2. Hypoxia-Associated SNPs Revealed Higher Lung Cancer Risks

Single nucleotide polymorphisms (SNPs) present on genes related to hypoxia signalling, such as VEGF, were investigated as risk factors and potential markers of lung cancer susceptibility. Interestingly, VEGF SNPs rs833061 (−460T > C), rs2010963 (+405C > G), and rs3025039 (936C > T) were frequently reported with positive associations to lung cancer but restricted to sub-groups based on tumour histology [52,213] or gender [214,215]. These variable results were pooled and analysed in a meta-analysis of 13 studies including a total of 8823 patients (healthy volunteers and lung cancer patients) and no association was observed for VEGF SNP rs3025039, rs833061, and rs699947 (−2578C/A) with risk of lung cancer [216]. Controversially, a meta-analysis focusing on VEGF SNP rs699947 described lung cancer susceptibility in both the Asian and overall populations [217]. HIF-associated SNPs were also explored in lung cancer susceptibility. HIF-1α SNP rs11549465 was associated with overall cancer risk including lung cancer risk, in a meta-analysis based on 26 383 patients, equally distributed between controls and cases [218]. In another meta-analysis, SNP rs11549465 was also associated with higher overall cancer risk among 16,440 patients [119]. Regarding ethnicity, breast and lung cancer were higher in the Asian population while rs11549465 was associated with a protective effect against lung and breast cancer in the European population. Such controversies and variable observations need to be validated, due to the impact of ethnicity [219,220]. Concerning HIF-2α, Wang et al. conducted a non-supervised analysis of multi-omic NSCLC datasets in order to explore the potential association with lung cancer risk [111]. As expected, several members of the HIF gene family were at the centre of the molecular network and more interestingly, EPAS1 (also known as HIF-2α) SNP rs12614710 was associated with lung adenocarcinoma.

#### 4.1.3. Hypoxia as a Prognostic Marker for Resected Tumours

Early NSCLC (represented by stages I and II) mostly benefit from surgical resections. Considering current guidelines, some sub-groups are eligible for post-operative chemotherapy based on platinum drugs despite a limited benefit on disease free-survival (DFS) and recurrence free survival (RFS) rates [221]. The identification of biomarkers able to predict poorer outcomes after a surgical resection could distinguish specific patient sub-groups that could benefit from adjuvant therapies. Thus, hypoxia was assessed as a potential biomarker in the surgical context.

##### HIF-1α Expression Is Associated with Worse Clinical Outcomes

HIF-1α was associated with poor patient outcomes after surgical resection for head and neck SqCC [222]. Based on multi-histological types of solid cancer, a large meta-analysis assessed HIF-1 and HIF-2 impacts on clinical outcomes including OS, DFS, RFS, progression-free survival (PFS), and cancer-specific survival (CSS). HIF-1α was associated with poor global outcomes, including NSCLC with no association in SqCC. Similar results with higher HIF-1α expression were observed in another meta-analysis on 2056 patients (pooled HR = 1.80, *p* = 0.003) [38]. However, such a meta-analysis relies on various studies that are not all consistent. For example, the expression levels of HIF-1α were assessed in a cohort of 66 NSCLC who underwent surgical resection [223], where high HIF-1α expression was associated with poor outcomes, but exclusively in node-negative patients and not in patients with lymph node metastasis. As hypothesised, hypoxia might play an important role in tumour growth and progression but not in established late stages.

##### Hypoxia Related Prognosis and HIFs

Other hypoxia-related markers were investigated as companions to predict clinical outcomes. For example, CA-IX or VEGF-A expressions were proposed as independent poor prognosis factors for RFS and OS [64,224,225]. VEGF overexpression was also investigated in a pooled meta-analysis of 7631 patients and compared to the prognosis. High VEGF-A expression was an independent poor prognosis factor, regardless of the histological sub-types (HR = 1.775, 95%CI: 1.384–2.275 for AC and HR = 2.919, 95%CI: 2.060–4.137 for SCC) [226]. VEGFs SNPs as a predictor of clinical outcomes in NSCLC were also investigated. However, results were heterogeneous with distinct proposed SNPs and contradictory observations [52,121,227]. HIF-2α relevance as a prognostic marker was investigated in a large meta-analysis containing various solid tumours [228]. A strong association with HIF-2α overexpression and poor clinical outcomes such as OS (HR = 1.69, 95%CI: 1.39–2.06), DFS (HR = 1.87, 95%CI: 1.2–2.92), RPS (HR = 2.67, 95%CI: 1.32–5.38), and PFS (HR = 2.18, 95%CI: 1.25–3.78), including in NSCLC were reported. Finally, non-invasive methods were also investigated such as PET, using hypoxic-dedicated tracers [^18^F]-FAZA in a cohort of localised NSCLC [229,230]. Tumours with hypoxic radiological features were associated with poorer PFS and OS [173,230,231].

Different approaches and markers related to hypoxia are available and potentially useful to predict clinical prognosis. However, additional studies are needed to select the most robust and relevant companion markers among them.

#### 4.1.4. Characterisation of Hypoxia to Improve the Clinical Course in Locally Advanced Stages

##### Current Strategy in Non-Resectable and Local NSCLC

About one-third of cases are diagnosed at a locally advanced stage, mostly only eligible to radiotherapy associated with platinum-based doublet chemotherapy and with poorer outcomes than surgically resectable tumours [232]. Clinical outcomes improved particularly with the use of durvalumab, an anti-PD-L1 immune blocker in maintenance. Nonetheless, the prognosis remained poor and many patients did not respond to radiotherapy and/or chemotherapy or developed major adverse-related events such as radiation-induced pneumonitis [233]. The need to identify markers able to predict clinical trajectories and to adapt cancer management individually is highlighted. Different strategies were adopted to investigate the impact of hypoxia in locally advanced stages of NSCLC.

##### Hypoxic Features to Predict and Monitor Tumour Response

Hypoxic-related markers through [^18^F]-MISO PET were assessed in 29 patients before their respective treatments [234]. Pre-therapeutic fractional hypoxic volume was thus associated with treatment response and PFS in an independent manner. Similar results were reported with [^18^F]-FAZA PET [235] and VEGF SNP rs833061 [236]. Additionally, hypoxia-related markers were monitored during the course of chemoradiation. A decrease in HIF-1α serum level was observed in concurrence with tumour reduction and potentially predicted treatment response (*p* < 0.001) [137]. [^18^F]-FAZA PET features at baseline were compared two and four weeks after treatment initiation and acquisition at the second week was proposed as the most relevant feature in the clinical management of NSCLC based on hypoxic parameters [237]. [^18^F]-FAZA PET evolution at a later post-treatment end-point was assessed without significant correlations with DFS [235]. Beyond the prediction of treatment response, hypoxia was also proposed as a potential safety-marker in patients treated by radiotherapy [238,239,240]. VEGFs SNPs could be predictors of radiation pneumonitis while others could be protectors [241]. Hypoxia was also associated with a higher risk of treatment-related deaths in patients undertaking chemoradiation [239]. As previously reviewed, hypoxia characterisation from early to locally advanced stages might improve NSCLC detection, staging, and selection of the best clinical strategy to enhance patient outcomes. Although many results coincide, the most relevant approaches need to be characterised.

### 4.2. Metastatic Stages: Potential Hypoxia-Related Treatment Strategies, from Response to Resistance

Contrary to early and locally advanced stages, many patients with metastatic NSCLC are not entitled to curative treatments. The objectives of the therapeutic strategy are to reduce and/or stabilise both loco-regional and distant disease and to prevent additional metastatic sites that burden OS. Current approaches combine various regimens of chemotherapy, immunotherapy, and targeted therapy such as EGFR-TKIs. Assuming that hypoxia was the main factor and a key player interacting with the progress of metastasis [242], our study investigated how hypoxic characterisation could enhance lung cancer management through each clinical therapeutic option currently available. Higher tumour incidence and maintenance of CSC or metabolic adaptation in hypoxic conditions were reported, suggesting that hypoxia may act as a major driver of resistance [243,244,245], with potential differential effects related to HIF-1α [246] and HIF-2 [247].

#### 4.2.1. Hypoxic Tumours Are Associated with a Higher Risk of Distant Metastasis

Considering distant visceral metastasis detection, hypoxia was broadly identified as a promoter of cancer growth and metastasis in various cancer types, including lung cancer [248]. More specifically, bone marrow metastasis, invasion, and colonisation were found associated with hypoxia [249]. HIF-1α expression in primary tumour tissues and the presence of bone metastasis was investigated in a retrospective cohort study of 96 patients with NSCLC [250]. Patients with bone metastasis harboured higher HIF-1α expression (in both intensity and tumour positive stained cell) than patients without bone metastasis. This positive association persisted in the adjusted analysis. The identification of tumour-harbouring hypoxic features could thus help clinicians evaluate the presence of bone marrow metastasis. These reports suggest that hypoxia (through HIF-1α or other related markers) could enhance NSCLC staging, both for lymph node and visceral metastasis.

#### 4.2.2. Chemotherapy

Among different drugs currently indicated in NSCLC, the platinum-based doublet regimen remains the main combination used, especially in first or second-line. However, response rates do not exceed 40% in first [251,252,253] and 15% in second-line. Hypoxia was investigated and proposed as a driver of chemoresistance in lung cancer [254]. A cohort of patients including 602 NSCLC treated by platinum-based regimen was genotyped on 42 genes involved in hypoxia sensing. Two variants (EXO1 (exonuclease 1) and RPA1 (replication protein A1)) were associated with worse response and clinical outcomes [255]. Complementarily, HIF-1α sequencing established a positive association between the presence of the 1772 CC genotype and higher response to platin salts and longer OS. In the same cohort, HIF-1α protein expression was a predictor of non-response to platinum-based chemotherapies [256]. Hypoxia was also evaluated in a non-invasive manner, by assessing VEGF serum level in patients treated with a first-line of carboplatin and paclitaxel at baseline and after two cycles of treatment. Whereas no significant changes in VEGF concentrations were noticed between the two-time points, responsive patients (including partial and stable disease) had lower levels of soluble VEGF than non-responders [257]. Other reports were consistent, based on lymphangiogenesis-related markers such as VEGF-C, VEGF-D, and soluble VEGF-R2 [258] or exploring other mono chemotherapy treatments employed in NSCLC such as paclitaxel or vinorelbine [259,260]. All those observations of hypoxic-driven chemoresistance in lung cancer are supported by preclinical reports and identified different actors, including direct involvement of HIF-1α, p53 mutation [261,262], CSC [263], YTHDF1 (YTH N6-methyladenosine RNA binding protein 1) [264], miR-128, and miR-155 [148]. Other non-specific mechanisms of resistance were described such as hypoxia-induced P-glycoprotein (p-Gp) [265] or epidermal growth factor-like domain 7, associated with multidrug resistance [266].

#### 4.2.3. Immunotherapy

##### Current Use of Immune-Checkpoint Inhibitors

Immunotherapies with a monoclonal antibody targeting anti-PD-L1 as a spearhead rapidly emerged and were placed in first-line treatments [251]. However, combined or used alone, response rates do not exceed 50% and relapse fatally occurs. As previously reported, hypoxia affects histological immune features and the tumour microenvironment by ultimately converging to an immune escape.

##### Hypoxia as a Predictor of Response under ICI Regimens

PD-L1 and VEGF-A were significantly co-expressed in a cohort of 129 surgical specimens (*p* = 0.002, r = −0.274), suggesting a potential interplay between hypoxia and PD-L1. Moreover, co-expressing sub-groups were associated with worse PFS and OS than the control group (*p* = 0.005), unveiling a potential interest in also targeting this association in the early stages [267]. This was supported by both preclinical and clinical models [268,269,270]. Interestingly, radiomics features were used to characterise hypoxic status in a cohort of 59 NSCLC treated by nivolumab [231] where tumours with hypoxic features had poorer clinical outcomes with anti-PD-1 use than non-hypoxic tumours. Immune resistance mediated by hypoxia is supported by preclinical experiments involving notch-hedgehog signalling and CSCs [271] Interestingly, therapeutic oxygenation may reverse this immune resistance [272,273]. However, only a few studies have investigated immune responses considering tumour hypoxic characterisation [274,275,276].

#### 4.2.4. Radiotherapy

Due to radiation-induced damage and its biological effects, hypoxia has been extensively investigated through radiation therapies in various solid cancers. PET imaging was mainly used for tumour hypoxic characterisation and enabled longitudinal monitoring during radiation [277].

##### Hypoxic Imaging to Guide Radiotherapy Strategies

Poor prognosis in patients with hypoxic tumours treated by radiotherapy was established across different radiation modalities [229,278,279,280,281]. It prompted to adapt, guide, and enhance radiotherapy intensity in hypoxic tumour sub-areas [282,283,284,285,286]. Thus, the RTEP-5 (radiotherapy dose complement in the treatment of hypoxic lesions) study experimented with a radiotherapy dose increase according to tumour-associated hypoxic features assessed by [^18^F]-MISO PET in unresectable NSCLC. Despite high dose escalation up to 86 Gy, this trial did not meet its first primary endpoint, whereas local radiotherapy intensification seemed to improve OS in its updated content with more frequent radiation-related adverse events [287]. Radiotherapy based on hypoxic-imaging characterisation appears insufficient and additional biomarkers are needed.

##### Biological Hypoxia-Related Markers Improving Radiotherapy Success

Several hypoxia gene signatures were developed as predictors of the radiotherapy response [142,288,289]. Radiation tumour resistance is complex [290] and involves many hypoxic elements such as HIF-1/2, ROS [291,292,293,294], heat shock proteins [295,296], or miR-210 complex [297,298]. They could emerge as promising hypoxia-related biomarkers to predict response and outcomes with radiotherapy, but clinical validation is required. Hypoxia-induced radio-resistance was modified by EGFR mutation status. EGFR mutant NSCLC cells were more sensitive to radiation than non-mutant cells which suggests a potentially relevant interplay between hypoxia and EGFR in NSCLC [299]. Hypoxic assessment as a guideline for radiotherapy strategies should be further evaluated alone or in combination with the genomic status or the immune profile.

#### 4.2.5. Targeted Therapies

##### KRAS

Kirsten rat sarcoma (KRAS) mutations remain the most common molecular alterations in NSCLC. Despite considerable efforts of investigation, this molecular pattern has only experimental therapies [300,301]. A COPD-like inflammation supported a tumourigenic effect of KRAS mutant lung cancer, driven by HIF-1α [302,303]. Additionally, KRAS mutant NSCLC secreted a specific cytokine profile with neutrophil recruitment into the tumour micro-environment, resulting in a pro-hypoxic condition [304]. HIF-1α inhibition (by Ras-PI3K-Akt pathway downregulation) in cell line models harbouring KRAS mutation repressed metastatic behaviour (proliferative and metastatic properties) [305]. Hypoxia consequences on KRAS mutant tumours have been investigated with other therapies. KRAS mutant cells were incubated with 3-bromopyruvate (3-BrPA), a promising alkylating metabolic inhibitor of cancer with KRAS mutation. While non-hypoxic cells exhibited a standard sensitivity, hypoxic conditions led to drug resistance for this molecule [306]. Thus, experimental data suggested that examining both KRAS molecular mutation and hypoxic status could have particular clinical significance. Considering the effectiveness of anti-KRAS therapies, selected patients with KRAS-mutant NSCLC harbouring non-hypoxic patterns could thus benefit from KRAS inhibitors.

##### EGFR

•Current Clinical Application of anti-EGFR in NSCLC

Several targeted therapies have been developed over the past decades to block activating EGFR mutations such as erlotinib, afatinib, and gefitinib. Those actionable mutations are observed in NSCLC sub-groups and represent about 10–15% of patients. Response rates and OS were drastically improved with these drugs. Osimertinib, an EGFR-TKI designed to overcome T790M—a common second mutation of resistance—is indicated and employed in first-line treatment. However, primary or secondary resistance inevitably occurs, and in many cases in the absence of additionally acquired mutations [307]. Interestingly, erlotinib is indicated in the global population and has demonstrated its efficacy in a non-mutated EGFR NSCLC population without an identified response or resistance-marker to date [308], highlighting EGFR-TKIs activity in non-mutant tumours, and the lack of relevant markers in this situation. We then reviewed the potential interest of defining hypoxia to improve EGFR-TKIs management, in both wild type (WT) and mutant EGFR NSCLC.

•Hypoxia Could Predict Non-Response to EGFR TKIs in Wild-Type EGFR NSCLC

In WT EGFR NSCLC, gefitinib and erlotinib treatment lead to a decrease in VEGF concentrations in tumour cells. This complex molecular mechanism involves HIF-1-dependent and independent pathways [309]. The interplay between EGFR resistance and hypoxia was investigated on NSCLC cells exposed to gefitinib and varying O2 levels. Interestingly, hypoxic NSCLC gefitinib-treated cells harboured greater resistance which was supported by IL-6 secretion and adaptive tumour inflammation [310]. Beyond EGFR mutation in a global population, HIF-1α expression was associated with poor outcomes, also supporting the need to consider a tandem for EGFR and hypoxia [311]. Moreover, 42 unselected patients with NSCLC were recruited in a pilot study to identify markers able to predict outcomes with combined anti-VEGF/anti-EGFR treatment in q WT-EGFR population. Gene expression profiling was performed before treatment with bevacizumab and erlotinib. Interestingly, gene set enrichment analysis identified hypoxia and angiogenesis-related signatures. Cluster analysis highlighted a hypoxia signature that distinguished two clusters based on prognosis. Patients with higher hypoxia-related gene expression were associated with lower response rates to combined therapies and poorer outcomes [312]. Both preclinical and clinical observations suggested that hypoxic features in EGFR-WT NSCLC could predict poor responses to EGFR-TKIs.

•Hypoxia Led to EGFR TKIs Resistance in Mutant EGFR NSCLC

Hypoxia-induced Gefitinib resistance in EGFR-mutant specimens is broadly mediated by HIF-1α [313,314]. Hypoxia seemed to support various mechanisms of resistance. For example, 202 NSCLC specimens were screened for EGFR mutations where mutant EGFR tumours were found to harbour higher expressions of MET. This pattern constituted a by-pass mechanism of resistance to EGFR-TKIs that was induced by HIF-1α [315,316]. Other mechanisms of resistance have been described, including the activation of insulin-like growth factor 1 receptor (IGFR1) [317], LSD-1 (lysine-specific histone demethylase 1) [318] or JUN [319]. Beyond first-line resistance to EGFR-TKIs, acquisition of T790M EGFR mutation is common. Higher HIF-1α gene expression in both chemo-resistant NSCLC and mesenchymal polarised tumours was observed [320]. In this context, HIF-1α overexpression led to EMT and chemoresistance while its inhibition restored chemosensitivity [321], highlighting a central role of hypoxia, even in tumours harbouring T790M mutation.

•Promising Strategies to Overcome Hypoxia-Mediated Resistance with EGFR-TKIs

In NSCLC harbouring exon 19 EGFR deletions, the erlotinib-cisplatin combination allowed to inhibit both HIF-1α and VEGF pathways, and was associated with greater responses [322], suggesting a potential interest in mutant-EGFR NSCLC with a resistance to EGFR-TKIs supported by hypoxia. Furthermore, an interim analysis of a randomised phase 3 trial assessing erlotinib/bevacizumab vs. erlotinib in 228 patients with a mutant EGFR status was published recently. Despite inconclusive results, combination drugs significantly improved clinical outcomes compared to erlotinib alone (HR: 0.605, 95%CI 0.417–0.877; *p* = 0.016), suggesting clinical relevance in targeting hypoxia for the EGFR mutation context, [323] and further trials are ongoing [324,325,326].

##### ALK

ALK-TKIs such as crizotinib or alectinib were developed to counteract constitutive activation of ALK receptor, mostly mediated by activating ALK fusions such as ALK-EML4 [327]. Unfortunately, clinical course is still conditioned to primary and inevitable secondary resistance. As observed in EGFR and hypoxia interplay, data suggest a close relationship between those ALK alterations and hypoxic condition. It was even described that ALK directly controlled HIF concentrations in hypoxic conditions for NSCLC [328]. Resistance to ALK-TKIs can be mediated by various mechanisms [329] where hypoxic conditions might play a potential role [330]. Thus, a hypoxia-mediated resistance to crizotinib via EMT induction was reported [331]. Interestingly, this resistance was labile and reversible after hypoxic reversion or HIF-1α inhibition, highlighting the potential relevance to counteract hypoxia in ALK-rearranged tumours harbouring TKI resistances. However, resistance to therapies involves many other factors such as PD-L1 overexpression mediated by HIF-1α in ALK-rearranged NSCLC specimens and promoting immune escape [332]. The development of hypoxia counteractors with ALK-related TKIs is a current challenge for future therapeutics [333].

## 5. Hypoxia-Related Treatments and Research Development

Current systemic approaches including chemotherapies, immunotherapies, TKIs, or radiotherapy are facing primary or secondary resistance where hypoxia can sustain a major role. As the tumour cancer cells and/or the tumour microenvironment may trigger hypoxia, several strategies have been designed to reverse its negative effects. In this section, we will review several therapeutic approaches targeting hypoxia: reduction of oxygen consumption, hypoxia-activated cytotoxic prodrugs, blockers of anaerobic metabolism, or anti-angiogenic molecules.

### 5.1. Pyruvate Dehydrogenase Kinase (PDK) Inhibitors

HIFs transcriptional complex largely interferes with glucose metabolism in hypoxic cells [334]. PDK genes are expressed and regulated by a HIF-associated transcription program via downregulation of mitochondrial oxygen consumption leading to a reduction of oxygen consumption [335]. PDK inhibitors have been designed to hijack this hypoxic tumour cell adaptation [336].

Among various PDK isoforms, PDK1 is featured as a key player in NSCLC metabolic adaptation [337]. Dichloroacetate (DCA) is a PDK1 inhibitor that was investigated in various solid cancers including NSCLC. Preclinical data on NSCLC cell lines were encouraging with anti-tumoural effect, synergistic with EGFR-TKIs (for erlotinib and gefitinib) [338], and overcome acquired-resistance to EGFR-TKIs [339]. However, no clinical trial was conducted in EGFR-mutant conditions to date. Excluding the EGFR-addiction context, DCA was tested in addition with 2-methoxyestradiol (2-ME), a metabolite reducing angiogenesis, replication, and HIF-1α pathway in human tumour cell-lines. The combination exhibited promising anti-tumour effects in preclinical A549 NSCLC cell lines [340]. Current clinical data available on DCA explored its anti-tumour properties in a phase II trial, including 6 patients with NSCLC (and 1 patient with breast cancer). Unfortunately, the clinical trial was closed regarding safety concerns after seven enrolments [341]. No additional trials are active concerning DCA, probably due to its worse tolerance profile.

The Cpd64 (2,2-Dichloro-1-(4-isopropoxy-3-nitrophenyl)ethan-1-one) was also proposed as a more specific and potent PDK1 inhibitor. It was tested in the human EGFR-mutant NSCLC cell line and demonstrated synergistic effects with erlotinib, especially by an increase of anti-proliferative effects under hypoxic conditions. In xenografts, Cpd64 increased the tumour growth inhibition in addition to erlotinib [342]. However, no clinical trial is investigating Cpd64 anti-tumour efficacity. Finally, PK-M2 appeared as specifically expressed in cancer cells and emerged as a promising target involved in the acquisition of metastatic and neoangiogenesis features [343,344,345]. However, no clinical data are currently available.

### 5.2. Metformin

Metformin, a well-known anti-diabetic drug has also been proposed as an anti-cancer treatment with immunogenic and anti-proliferative properties, inhibiting oxygen consumption by both adenosine monophosphate-activated protein kinase (AMPK) dependant and independent pathways [346]. Metformin was thus explored in a small effective trial of 15 patients treated by radiotherapy. Modification of tumour activity evaluated by PET was noticed while the small effective limited robust conclusions [347]. Other clinical trials observed promising activity with metformin administration in combination with ICI for various solid cancers, including NSCLC [348]. To date, only retrospective studies are available regarding metformin in NSCLC. For example, in a controlled cohort of 100 patients, improved clinical outcomes were reported (ORR and DCR at 41% (*p* = 0.4) and 70% (*p* = 0.5), respectively) [349]. Another retrospective study of 166 patients treated by chemoradiotherapy reported a non-significate difference in combination with metformin [350]. Similar non-significate results were published in smaller cohorts [351,352]. A larger non-randomised retrospective study on nationwide prescription cohort proposed benefits associated with metformin co-administration in NSCLC patients [353]. Therefore, the administration of metformin in NSCLC is still under investigation (NCT02109549) and its place is being explored in various combinations such as immunotherapy (NCT02186847), or stereotaxic ablative radiation (NCT04170959), but not EGFR-TKI because of a negative randomised study of 224 patients with EGFR mutated NSCLC treated by gefitinib with or without metformin [354].

Other molecules influencing glucose metabolism in hypoxic conditions such as GLUT-1 were also explored. This HIF-1α partner has been mainly investigated as a prognostic marker in NSCLC [355]. Patient-derived xenograft models, NSCLC cell lines, primary human NSCLC cultures, and public Tumour Cancer Genome Atlas databases highlighted an important implication of GLUT-1 in NSCLC [356]. However, no clinical trial data are available in the context of lung cancer management.

### 5.3. Vorinostat

Vorinostat is a histone deacetylase (HDAC) inhibitor increasing HIF-1α degradation [357,358]. As its efficacy was insufficient in monotherapy [359,360,361], vorinostat was administered in a phase II randomised double-blind and placebo-controlled trial, associated with carboplatin/paclitaxel chemotherapy in 94 patients with NSCLC [362]. A response rate of 34% was observed in the vorinostat arm (vs. 12.5%, *p* = 0.02 in the placebo arm). The small effective did not permit to conclude on PFS and OS while a trend was observed in favour to vorinostat combination (6.0 months vs. 4.1 months and 13.0 months vs. 9.7 months, respectively). The safety profile remained manageable, mainly represented by grade 4 thrombopenia which occurred in 18% vs. 3% in the placebo arm. Concurrently with stereotactic radiosurgery, a phase I dose-escalation trial was also conducted to determine a maximum tolerated dose for vorinostat in monotherapy as a radiosensitiser in 14 NSCLC patients [363]. Although no dose-limiting toxicity was observed, five patients early withdrew the protocol, highlighting a labile clinical tolerance. Concerning EGFR mutation, a non-randomised phase I/II trial enrolled 33 NSCLC patients experiencing resistance to erlotinib. Vorinostat (400 mg/day) was added to erlotinib and PFS was evaluated at 12 weeks. Vorinostat was insufficient to reverse EGFR-TKI acquired resistance in EGFR mutant condition [364]. Similarly, vorinostat failed to improve PFS in combination with gefitinib in other clinical trials including patients with EGFR-mutant NSCLC [365,366]. ICI and particularly pembrolizumab combined with vorinostat (400 mg/day) was explored in a phase 1 study, including 33 patients with advanced NSCLC. No additional safety concerns were reported and a disease control rate of 67% was reported (including four (13%) partial response and 16 (53%) stable disease) [367]. Such promising preliminary results justified current recruitment for Pembrolizumab combined with vorinostat in metastatic NSCLC (NCT02638090).

### 5.4. Nitroglycerin

Nitroglycerin (NTG) is a vasodilator molecule that inhibits HIF-1α, resulting in tumour blood flow increase and reduced oxygen privation. Its addition, it was evaluated in combination with various chemotherapy regimen for patients with NSCLC. A randomised phase II trial enrolled 120 patients harbouring advanced NSCLC and evaluated the addition of NTG patches (25 mg daily for 5 days) to the vinorelbine/cisplatin chemotherapy regimen [368]. This trial was positive, with a response rate increased to 72% in the intervention NTG-arm vs. 42% in the control group. Both PFS and OF were significantly improved (adjusted HR_PFS_ = 2.1, *p* = 0.002 and adjusted HR_OS_ = 2.5, *p* < 0.001 respectively). A randomised phase II trial enrolled 223 chemo-naïve patients with metastatic non-squamous NSCLC, comparing carboplatin-paclitaxel-bevacizumab with or without NTG patches (15 mg for 4 days per cycle) [369]. The safety profile was noticed while no improvement of PFS or OS was reported in favour of the NTG arm. Such discordance between the two trials might be attributed to the bevacizumab as an anti-angiogenic drug. Although side effects remained manageable (mainly represented by headache, commonly observed with vasodilator treatments), a dedicated phase III trial investigating NTG addition for 5 days with conventional first-line chemotherapy in advanced NSCLC patients was stopped at first interim analysis for non-significance [370]. Its combination with radiotherapy failed to demonstrate a clinical interest and no additional investigations are currently evaluated for this approach [371].

### 5.5. Tirapazamine

As a model of a specific hypoxia-related drug, tirapazamine is a synthetic molecule that is activated in hypoxic conditions to a toxic radical [372]. This drug was thus investigated in a randomised 1:1 phase III of 446 patients with NSCLC, in combination with cisplatin. Both OS and response rate were improved (34.6 vs. 27.7 weeks, *p* = 0.078 and 27.5 vs. 13.7, *p* < 0.001; respectively) [373]. Despite positive results supporting clinical relevance, the development of tirapazamine in NSCLC was stopped without integration in clinical lung cancer management (NCT00066742), and no results were published.

### 5.6. Efaproxiral

As a fibric acid derivative displacing oxygen from hemoglobin to hypoxic tissue ultimately leading to an increase of oxygen delivery, efaproxiral exhibited promising results in phase I/II but failed in the later phase of clinical development, especially as a radiosensitiser and hypoxic target therapies [374,375,376]. Several additional clinical trials are ongoing with novel radiosensitisers [377,378], and modified radiotherapy protocols according to the hypoxic region to enhance immune cytogenic anti-cancer effect [379,380].

### 5.7. Anti-Angiogenic Therapies

They are currently available in lung cancer including bevacizumab, a monoclonal anti-VEGF antibody [381]. Associated with an improvement in clinical outcomes, it remains restricted to sub-populations and excluded from regular cases such as SqCC, brain metastasis, or thrombotic history [382]. Despite the development of bevacizumab as a drug specifically targeting neoangiogenesis, no predictive markers for angiogenic inhibition are available. To date, bevacizumab is still indicated in combination with other treatments, especially chemotherapies [383]. Other anti-angiogenic drugs are mostly investigated in combination with other treatments such as immunotherapy or targeted therapy, based on potential synergistic efficacy [384].

More conventional anti-angiogenic therapies such as bevacizumab gain interest [385] and both clinical and preclinical data are cumulating for synergistic effects of angiogenic inhibition with ICI [386]. Dedicated clinical trials are ongoing, questioning the addition of anti-angiogenic treatment to immunotherapy and chemotherapy now widely used in the NSCLC first-line for pembrolizumab [387] or atezolizumab (NCT03836066, NCT03896074). Interestingly, lenvatinib (a multi-receptor TKI targeting VEGFRs and FGFRs) exhibited both safety and signal of activity in combination with pembrolizumab or chemotherapy in phases Ib/II [388,389]. Several trials are ongoing for lenvatinib in combination with chemo- and immunotherapies specifically in NSCLC (NCT03829332; NCT03829319; NCT04875585) and should provide their results in the coming years.

## 6. Conclusions

In this study, we conducted an extensive review of the potential impact of hypoxia in each stage of NSCLC and highlighted clinical and pathological features related to hypoxic tumours relevant to clinicians. It appears that some tumours are more frequently associated with hypoxic regions than others, such as poorly differentiated SqCC presenting a tumour microenvironment including high tumoural microvessel density and stroma-enriched immune cells harbouring epithelial-to-mesenchymal polarisation. The current challenge to identifying hypoxia remains the definition of relevant thresholds for markers discriminating hypoxic from non-hypoxic tumours. Pathological examinations and immunostainings are needed to validate further markers in paired and matched comparison studies. The identification of biological signatures based on nucleic acid expressions may contribute to the development of hypoxic-scores after validation in larger and design-dedicated cohorts. Systematic genetic association studies taking hypoxia as a relevant parameter will ideally complement the translational approach. We also reviewed current promising approaches allowing to evaluate hypoxia in the NSCLC context with a special interest in the most suitable and transposable approaches in clinical routine. Tumour hypoxia biology is complex and in constant evolution over time, given that sequential drugs and radiation use lead to resistance and treatment escape. We finally investigated how hypoxic characterisation could influence the major steps of lung cancer clinical management. For patients with early cured NSCLC, transversal hypoxic tumour determination might also be of interest to isolate and predict those who would benefit from adjuvant therapies to reduce the risk of relapse. Nonetheless, challenges and clinical goals are specific in advanced and metastatic stages when aiming to predict tumour response across various regimens of treatments. In these later stages, longitudinal hypoxic characterisation might be the most relevant approach. Repeated PET/CT scans and more strikingly, circulating hypoxia-related markers may enable monitoring of tumour variations and adaptation of clinical strategies in a personalised approach. Despite limitations to hypoxia implementation in lung cancer clinical management, evidence is accumulating for its consideration, including dedicated hypoxia-related treatments. Hypoxia characterisation could improve the outcome of patients with NSCLC and might represent the next step to a personalised medical protocol in the field of cancer.

## Figures and Tables

**Figure 1 cancers-13-03421-f001:**
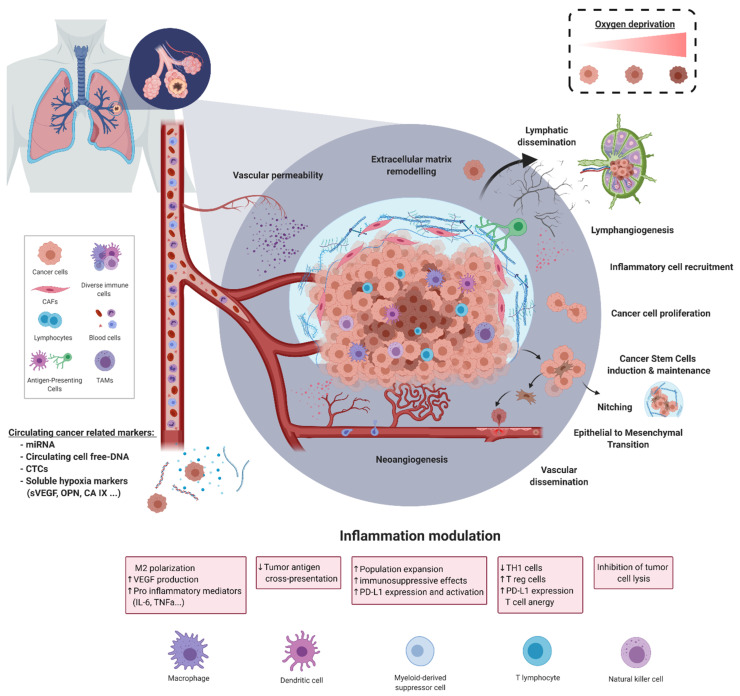
Schematic representation of lung cancer properties induced by hypoxic conditions. Hypoxia drives many mechanisms broadly involved in aggressiveness, invasion, and leading to the acquisition of metastatic properties. Additionally, oxygen deprivation supports the modulation of inflammation associated with an immune escape.

**Figure 2 cancers-13-03421-f002:**
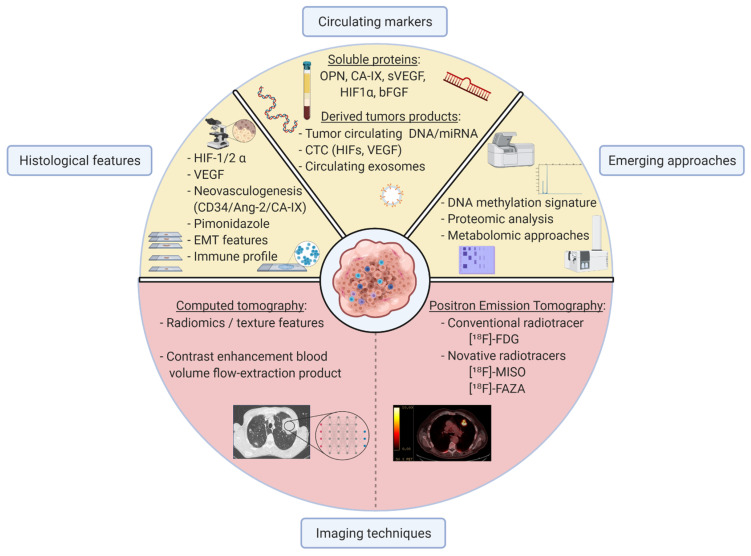
Molecular, cellular, and whole-organism techniques to assess hypoxic status in lung cancer. Various methods to define hypoxic features are currently available or in development to meet clinician’s expectation.

**Table 1 cancers-13-03421-t001:** Hypoxia-related markers with a clinical interest in non-small cell lung cancer. HIF-1 α: hypoxia-inducible. Figure 1. alpha, OS: overall survival, HIF-2 α: hypoxia-inducible factor-2 alpha, Ang-2: angiopoietin-2, GLUT-1: glucose transporter-1, CA-IX: carbonic anhydrase-IX, OPN: osteopontin, VEGF: vascular endothelial growth factor, bFGF: basic fibroblast growth factor, CCL20: C-C motif chemokine ligand 20, CORO1C: coronin-1C, CTSC: cathepsin C, LDHA: lactate deshydrogenase A, NDRG: N-Myc downstream regulated 1, PTP4A3: protein tyrosine phosphatase 4A3, TUBA1B: tubulin alpha.

Hypoxic Marker	Biological Material	Method of Assessment	Observation	Clinical Interest	Advantage	Limitation	Patients (*n*)	REF
HIF-1 α	Tumour cells	Histological	↑ expression associated with lymph node metastasis	Staging	Reproducible	Variability on HIF-1α threshold of positivity and its intracellular localization	1436	[38]
	Tumour cells		↑ expression associated with ↓ OS	Prognosis			1049	[38]
							1113	[47]
							256	[136]
	Serum concentrations	Circulating blood marker	↓ serum concentration during chemoradiotherapy course	Response monitoring	Mini-invasive procedure Reproducible Real-time monitoring	No control group	80	[137]
HIF-2 α	Tumour cells	Histological	↑ expression associated with ↑ stages ↑ expression associated with ↓ OS	Staging Prognosis	Reproducible	Restricted to stages I to III No association with HIF-1 α expression	140	[50]
Ang-2	Tumour cells	Histological and mRNA expression	↑ expression associated with ↑stages ↑ expression associated with ↓ OS	Staging Prognosis	Reproducible	mRNA expression not currently transposable in clinical routine Significance restricted to AC subgroup	1244	[46]
	Serum concentration	Circulating blood marker	↑ concentration associated with lung cancer (Se 92.5 and Sp 97.5%)	Diagnosis	Mini-invasive procedure Reproducible Real-time monitoring	High threshold	228	[43]
			↑ expression associated with ↑ stages ↑ expression associated with ↓ OS	Staging Prognosis		-	575	[44]
GLUT-1	Tumour cells	Protein and mRNA expression	Co-expression with PD-L1 ↓ co- expression associated with ↑ OS	Prognosis	Reproducible	No evaluation of ICI sensibility	295	[104]
CA-IX	Serum concentration	Circulating blood marker	↑ concentrations associated with ↓ survival rates under radiotherapy	Prognosis	Mini-invasive procedure Reproducible Real-time monitoring	Small effective	55	[138]
Flow extraction product	CT radiomic analysis	CT features	↓ enhancement associated with lymph node metastasis Correlation with GLUT expression	Staging	Radiometabolic hypoxia-related markers	Small effective Non-standardised techniques of acquisition and analyses	14	[139]
Standardised Uptake Values (SUV)	PET radiomic analysis	Metabolic features	Correlation with histological HIF-1α expression	Not established	Non-invasive procedure Tumour heterogeneity consideration	No investigation on prognosis	288	[140]
OPN	Serum concentration	Circulating blood marker	↑ concentrations associated with ↓ OS	Predictive of ↓ outcomes with chemoradiotherapy Predictive of ↓ outcomes with radiotherapy Predictive of tumoural response and prognosis with chemoradiotherapy	Non-invasive procedure Tumour heterogeneity consideration Reproducible Real-time monitoring	Included in a multiparametric model with other proteins	263	[141]
						Small effective	44	[142]
							81	[143]
							55	[138]
						Significance restricted to SqCC subgroup	337	[144]
VEGF	Tumour cells	Histological and mRNA expression	↑ concentration associated with ↓ OS	Monitoring of tumoural response under chemoradiotherapy	Reproducible	Variability on VEGF threshold of positivity and intracellular localization	1549	[145]
	Serum concentration	Circulating blood marker	↑ concentration associated with ↓ OS	Predictive of response and prognosis with immunotherapy	Non-invasive procedure Tumour heterogeneity consideration Reproducible Real-time monitoring	Significance restricted to elderly (>75 y.o) or PS > 2.	235	[146]
			↑ concentration associated with ↓ survival rates under radiotherapy	Predictive of ↓ outcomes with radiotherapy		Small effective	55	[138]
bFGF	Serum concentration	Circulating blood marker	↑ concentration associated with ↓ OS	Monitoring of tumoural response under chemoradiotherapy regimen	Non-invasive procedure Tumour heterogeneity consideration Reproducible Real-time monitoring	Variability on bFGF threshold of positivity	358	[147]
miR-21, miR-128, miR-155, miR-181a	mi-RNA serum concentration	Circulating blood marker	Prediction of outcomes with first-line chemotherapy	Predictive of tumoural response Prognosis	Non-invasive procedure Reproducible Real-time monitoring	Not currently transposable in clinical routine Significance restricted to SqCC subgroup	128	[148]
CA-IX, CCL20, CORO1C, CTSC, LDHA, NDRG, PTP4A3, TUBA1B	Gene expression signature	mRNA expression	Predictive of hypoxic tumours associated with ↓ OS Co-expression with TILs infiltration	Prognosis	Non-invasive procedure Reproducible Real-time monitoring	Not currently transposable in clinical routine	515	[97]

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
