# Peer review of "Hypoxia in Lung Cancer Management: A Translational Approach"

_cancers, 2021, doi:10.3390/cancers13143421_

Round 1

Reviewer 1 Report

The review on Hypoxia in lung cancer management: a translational approach, is an extensive effort in which the authors present a huge number of publications (345 references) in the cancer field that generally support the concept that tumour hypoxia is associated with decreased progression-free survival and overall survival. The strengths of this manuscript lie in the extent of the review, the discussion of the potentially predictive value of hypoxia for survival, the biological consequences of hypoxia (figure 1) and the methods to detect tumor hypoxia (figure 2). Table 1 is also of interest as it highlights some of the hypoxia-related marker of interest that have been studied in NSCLC. However, this reviewer has several major issues with the very long and sometimes rambling presentation and an abundance of speculation on the translational and therapeutic implications of hypoxia in lung cancer care.

Major Comments

  1. Abstract: “Hypoxia could support screening programs and predict patients with higher risks of NSCLC…” There is no clinically credible evidence in support of this statement at all. Also, I could find no support for the conclusion that “hypoxia might also guide the choice of regimen in advanced stages.” Such strong statements in a review article on lung cancer need to be backed by clinical study evidence.
  2. Page 2, lines 75-77: The authors attempt to link “alveolar gas exchange and haematosis”(?) with the particular relevance of hypoxia in lung cancer. There is no relevance. Most lung cancers do not originate in the alveoli but in the airways. But even if lung cancer originated in the alveoli (which it mostly does not) this still would not mean that hypoxia is more relevant in the lung.
  3. The organization of the paper’s sections is not clear. Other than an Introduction and a conclusion, the authors divide the paper into 3 sections: Identification of hypoxia in tumors, Hypoxia characterization in clinical conditions and patient selection and Hypoxia as a companion biomarker for clinicians. The titles do not really differ from one another. Why not structure the paper with meaningful sections such as they have done in the figures and tables: For example: 1. Biological consequences of hypoxia 2. Tools to detect hypoxia in patients 3. Prognostic implications of hypoxia in lung cancer, 4. Potential hypoxia-related drug targets 5. Future research.
  4. There is some unnecessary repetition in the text. For example, serum markers of hypoxia are discussed on page 11 and again on page 15.
  5. All comments related to using hypoxia markers as predictive tools for lung cancer screening should be removed (abstract, pages 14 and 15, conclusion page 21). There is no evidence that this is a viable or even a promising approach to screen individuals for lung cancer.
  6. Figure 3 should be removed. This is at best a purely speculative proposal to base therapies for lung cancer on hypoxia markers, and at worst could be dangerous. Any changes in patient care must be based on well-designed clinical trials. No such trials exist in support of this therapeutic proposal.
  7. The authors discuss using “hypoxia inhibitors” – this is very vague. At one point, Erlotinib and cisplatin are said to be HIF-1a inhibitors (they do a lot more than inhibit hypoxia!) but little else is suggested as a hypoxia-inhibitor. Why not discuss other more specific inhibitors of the hypoxia-dependent signaling pathway. For example, the PDK inhibitor, dichloroacetate, PK-M2 stabilizers, GLUT-1 inhibitors and HIF-1a inhibitors. If this is to be done, a section on the cellular metabolic consequences of hypoxia may be needed (aerobic glycolysis).
  8. The section on page 10 entitled “Current limitations for hypoxia characterization by PET” does not discuss the limitations of 18FDG-PET scanning to quantify hypoxia in lung cancer. The review presents several arguments in favor of 18FDG-PET which is widely available and has broad implementation in cancer clinics. What are its limitations?

Minor points

  1. Although the English syntax is generally good, some areas need improvement. Abstract: “…in presence of…” Also, page 2 line 87: “The expression of… nitroimidazoles is triggered by…” Nitroimidazoles are not genes or proteins that are expressed or triggered.
  2. The reference to HIF-1a as a marker for NSCLC dates from 1992 and has never resulted in a large follow-up study as the authors propose. Is there any more recent evidence for this marker in lung cancer, and if not, why not?
  3. Figure 2: “Pimomidazole” should be corrected to “pimonidazole” in the figure.
  4. Table 1 and all through the text, OS needs to be defined at its first occurrence.
  5. Table 1, page 8: GLUT1 – the arrow indicates that expression of GLUT-1 is associated with better OS. The reference (#104) indicates the opposite.
  6. Page 18, lines 689-691. What are the “drugs targeting hypoxia-related pathways”? In the reference 284, the drugs are chemotherapy agents that are not specific to hypoxia pathways and the NCT02992912 study that is referenced has nothing to do with hypoxia-targeting drugs.
  7. Page 19, lines749-751, the authors state that “We explored the potential interest of defining hypoxia EGFR-TKIs management, in both wild type (WT) and mutant EGFR NSCLC.” Do they mean they did this study – and if so where is the reference? Alternatively, is this simply a way of making the text transition to the next paragraph? Ambiguous. Please clarify.

Reviewer 2 Report

The authors of the manuscript "Hypoxia in Lung Cancer Management: A Translational Approach" reported emerging data highlighting the role of hypoxia as a biomarker in lung cancer management and therapy. This review is very interesting and full of information that considers hypoxia characterization as the next step in personalized cancer medicine protocols.

In lane 763, the authors wrote: Whole genome sequencing” but then they describe data of gene expression

Reviewer 3 Report

The authors describe this review paper on hypoxia in lung cancer management. Lung cancer is the number one cause of death from cancer worldwide and remains a challenging public health problem. Overall the description is well written. However, the manuscript as it currently stands raises some questions that need to be addressed.

Minor comments:

In 'Detecting hypoxia-inducing factors in whole tumor tissue' (section 2.1.), the authors describe HIF-1alpha, VEGF, CA-IX, etc. as major factors for detecting hypoxia. As you know, VEGF, CA-IX and Glut1 are target genes of HIF. Therefore, an explanation of HIF and its target genes is required.
